# Teacher-Guided Graph Contrastive Learning

**Jay Nandy**[*]                                                        *jayjaynandy@gmail.com*
*Fujitsu Research India Private Limited*

**Arnab Kumar Mondal**[*]                                  *arnabkumarmondal123@gmail.com*
*Fujitsu Research India Private Limited*

**Manohar Kaul**[†]                                              *Manohar.Kaul@fujitsu.com*
*Fujitsu Research India Private Limited*

**Prathosh AP**                                                          *prathosh@iisc.ac.in*
*Fujitsu Research India Private Limited*
*Indian Institute of Science Bengaluru*

**Reviewed on OpenReview:** *https://openreview.net/forum?id=15tjpSHI15*

## Abstract

State-of-the-art self-supervised representation learning methods for Graphs are typically based on contrastive learning (CL) principles. These CL objective functions can be posed as a supervised discriminative task using *'hard'* labels that consider any minor augmented pairs of graphs as 'equally positive'. However, such a notion of 'equal' pairs is incorrect for graphs as even a smaller 'discrete' perturbation may lead to large semantic changes that should be carefully encapsulated within the learned representations. This paper proposes a novel CL framework for GNNs, called *Teacher-guided Graph Contrastive Learning (TGCL)*, that incorporates 'soft' pseudo-labels to facilitate a more regularized discrimination. In particular, we propose a teacher-student framework where the student learns the representation by distilling the teacher's perception. Our TGCL framework can be adapted to existing CL methods to enhance their performance. Our empirical findings validate these claims on both inductive and transductive settings across diverse downstream tasks, including molecular graphs and social networks. Our experiments on benchmark datasets demonstrate that our framework consistently improves the average AUROC scores for molecules' property prediction and social network link prediction. Our code is available at https://github.com/jayjaynandy/TGCL.

## 1 Introduction

Graphs are versatile data structures representing relationships between entities in various domains, such as social networks (Ohtsuki et al., 2006; Fan et al., 2019), bio-informatics (Muzio et al., 2021), and knowledge graphs (Wang et al., 2014; Baek et al., 2020). Analyzing and understanding graph data is crucial in many real-world applications, including community detection (Fortunato, 2010), node classification (Bhagat et al., 2011), link prediction (Zhang & Chen, 2018; Rossi et al., 2021), recommendation (Wu et al., 2019), continual learning (Mondal et al., 2024), and time-series analysis (Chauhan et al., 2022). Graph representation learning can potentially make significant leaps in graph-based analysis and prediction.

*Self-supervised Learning (SSL)* for graphs has emerged as an important research area that leverages the inherent structure or content of inputs to learn informative representations without relying on explicit labels (Hu et al., 2020a; Hwang et al., 2020). Existing graph-SSL methods can be broadly categorized as:

---

[*]Equal contribution.
[†]Corresponding author

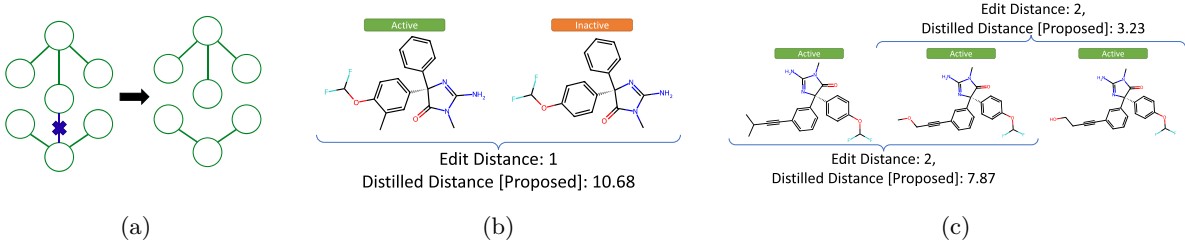

Figure 1: Illustrating the shortcomings of existing CL methods: **(a)** Even a 'minor change,' *i.e.*, removal of one edge can significantly change a graph's semantics, leading to disconnected components that are not captured using edit-distance-based discrepancy (Kim et al., 2022). **(b & c)** A more specific example of correlated-structured molecules that either actively bind to a set of human $\beta$-secretase inhibitors or inactive (Wu et al., 2018). **(b)** Molecules having dissimilar properties can have smaller edit distances, while **(c)** molecules from the same class can have larger edit distances. In other words, edit distance remains ineffective in capturing chemical semantics. Our proposed *distilled perception distance* form a pre-trained teacher incorporates 'soft' semantic distances for any arbitrary graphs to train a better student representation model.

**(a)** *local similarity-based predictive learning* & **(b)** *global similarity-based contrastive learning. Predictive learning*-based methods (Hu et al., 2020a; Kim & Oh, 2021; Rong et al., 2020) produces artificial labels by capturing specific local contextual information of neighborhood sub-graphical features to produce the representations. However, it restricts them to capturing only the local graph semantics. Alternatively, *contrastive learning (CL)*-based models for graphs aim to maximize the agreements between instances perturbed by *semantic-invariant augmentations* (positive views) while repelling the others (negative views) to capture global semantics. CL-based SSL models are extremely popular in the computer-vision community. For such applications, we can easily generate such semantic-invariant perturbations using simple techniques *e.g.,* rotation, flipping, and color jittering (Chen et al., 2020a; Grill et al., 2020b).

Several graph contrastive learning methods are also proposed where the positive pairs are produced using transformations *e.g.,* edge perturbation, attribute masking, and subgraph sampling. However, unlike *continuous* computer vision domains, even 'minor' modifications in the graph structures, such as removing one edge or node, can significantly change the properties of graphs due to their discrete nature (Figure 1a & 1b). Recently, (Kim et al., 2022) introduced *discrepancy-based self-supervised learning (D-SLA)* by incorporating *edit distance*-based discrepancy measures between two graphs to address these limitations. However, computing the edit distance between two arbitrary graphs is *NP-hard* (Sanfeliu & Fu, 1983; Zeng et al., 2009). Further, it can only provide high-level structural information without capturing any semantic differences (Figure 1a & 1b). In this paper, we propose a graph representation learning framework by incorporating more *semantically-rich* soft-discriminative features using such an imperfect pre-trained teacher to regularize the learning.

## 1.1 Motivation & Contributions

The existing CL methods for graphs can be viewed under the same umbrella where these techniques learn representations by contrasting different views of the input graphs. In principle, their loss functions can be considered as supervised classification objectives by creating pseudo-labels among different views of input graphs (Oord et al., 2018; Gutmann & Hyvärinen, 2010). In contrast, in the supervised learning literature, it has been observed that incorporating *'soft labels'*, even from an imperfect teacher, in the form of *Knowledge Distillation (KD)* leads to better generalization (Hinton et al., 2015; Menon et al., 2021; Kaplun et al., 2022). Given these prior results, we explore the following question: *Can 'soft' guidance from an imperfect teacher lead to a better CL framework for graphs?*

The fundamental idea of KD is to use softened labels via a teacher network while minimizing the supervised risk of a student network by reducing the divergence between their logits (Hinton et al., 2015). Prior works have shown that *Bayes-distilled risk has lower variance* compared to naive undistilled counterpart, which

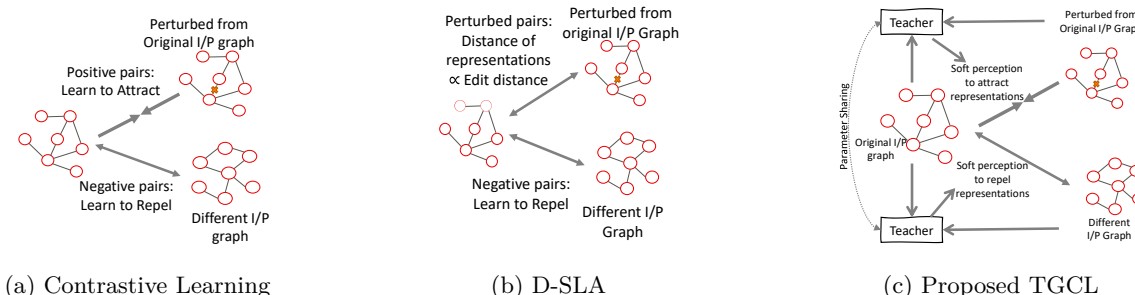

| (a) Contrastive Learning | (b) D-SLA | (c) Proposed TGCL |

Figure 2: Comparing our proposed TGCL framework with the existing contrastive learning You et al. (2020); Xu et al. (2021), and D-SLA Kim et al. (2022). From the classification point of view, the standard CL methods consider the similarity between the anchor and the perturbed graphs as "hard" positive pairs while the other graphs as "hard" negative pairs. D-SLA introduces "hard" discrepancies using edit distance between the anchor and the perturbed graphs, while the other graphs as "hard" negative pairs. Our proposed TGCL introduces a novel distilled perception distance for smooth discrimination between arbitrary graphs.

leads to better generalization (Menon et al., 2021). Motivated by these results, we propose a novel *Teacher-guided Graph Contrastive Learning (TGCL)* framework. We design a *distilled perception distance* (or *distilled distance*) between two arbitrary input graphs using their deep features obtained from a pre-trained "teacher" to define a softer notion of positive/negative pairs. We train the student network by incorporating such 'soft labels' for each pair of graphs. We argue that by introducing distilled distance, we can introduce the regularized semantic difference between two arbitrary graphs, addressing the shortcomings of the existing CL frameworks for graphs. For example, Figure 1c demonstrates that our distilled distance obtained from the "teacher" can significantly differ among molecular graphs with correlated structures towards capturing the chemical semantic differences for graphs. Figure 1b shows that the distilled distance captures the chemical semantic difference of molecules with different chemical properties, however, with a minor structural difference. The contributions of our work can be summarized as follows:

**1.** we propose to obtain distilled perceptual distances by comparing the deep features from a pre-trained teacher, followed by injecting them as "soft pseudo-labels" into the contrastive loss objective to appropriately capture the semantic differences between two arbitrary graphs in the student's representation space. Theoretically, by viewing the contrastive loss objective for graphs from a supervised loss, incorporating such 'distilled perceptual distances' acts as soft pseudo-labels that reduce the variance of Bayes-distilled risk to provide better graph representations. To the best of our knowledge, we are the first to propose such a teacher-guided soft-discrimination-based contrastive learning framework for the discrete domain of graphs.

**2.** Our proposed concept of 'soft-labeled' pairs of graphs can be adapted to any contrastive learning framework. We demonstrated two variations of TGCL frameworks by modifying the well-known NT-Xent loss Chen et al. (2020a); You et al. (2020) and D-SLA method Kim et al. (2022) to incorporate smooth perception from a teacher network for training the student network. Notably, TGCL is specifically designed for graphs to appropriately incorporate the representational distance even when minor perturbations significantly change the input semantics.

**3.** Experiments on graph classification for molecular datasets and link prediction on social network datasets where our proposed framework consistently outperforms the existing methods by improving upon the teacher. we improve the average *area under receiver operating curve (AUROC)* score by $\approx 2.23\%$ and $\approx 6\%$. for molecules' property prediction and social network link prediction tasks respectively.

## 2 Related Work

### 2.1 Representation Learning on Graphs

**Classical Approaches:** One of the most straightforward representations for a graph is to consider the *'bag of nodes'*. Weisfeiler-Lehman kernel (Shervashidze et al., 2011) improves upon this idea by utilizing an iterative

neighborhood aggregation strategy. One may also count the occurrence of small subgraph structures, called *graphlets*. However, it is a combinatorially challenging problem, and approximate algorithms are required (Ahmed et al., 2015; Hočevar & Demšar, 2014). A few other approaches enumerate different kinds of paths in graphs (Kashima et al., 2003; Borgwardt & Kriegel, 2005).

**Shallow Algorithms:** DeepWalk (Perozzi et al., 2014) and LINE (Tang et al., 2015) are random walk-based techniques using depth-first search (DFS) and breadth-first search (BFS), respectively. "node2vec" (Grover & Leskovec, 2016) combines both BFS and DFS to learn node embeddings by maximizing the likelihood of preserving node neighborhoods.

**Predictive SSL for Graphs:** These methods aim to predict specific graph properties, *e.g.,* predicting the attributes of masked nodes/edges (Hu et al., 2020a), existence of an edge (Hwang et al., 2020) or contextual properties and presence of motifs (Hu et al., 2020b; Rong et al., 2020). These predictive tasks serve as self-supervisions, as they do not require explicit supervised labels. Instead, they rely on the local sub-structure of the graph for producing labels.

**Contrastive SSL for Graphs:** Deep Graph Infomax (DGI) (Veličković et al., 2019) maximizes the mutual information between graph representation and patch representation. InfoGraph (Sun et al., 2020) maximizes the mutual information between the graph-level representation and the representations of substructures of different scales, such as nodes, edges, and triangles. Several other works (You et al., 2020; 2021; Zhu et al., 2021; Yin et al., 2022; Wang et al., 2022) employ contrastive learning by generating perturbed views of the original graph through attribute masking, edge perturbation, and subgraph sampling to obtain better representations. Recent works (S et al., 2021; Yang et al., 2021) also explore adversarial augmentation strategies to further improve these frameworks.

## 2.2 Knowledge Distillation (KD)

KD (Hinton et al., 2015) was originally introduced to transfer knowledge from a complex 'teacher' model with large capacity to an efficient 'student' model with lower capacity while performing similarly to the teacher. Several works also focus on improving the student's performance on a wide range of applications (Heo et al., 2019; Furlanello et al., 2018; Lopes et al., 2017; Li et al., 2021; Lee et al., 2018; Bhat et al., 2021).

**Surpassing the Teacher's performance.** KD allows the student to learn from both the raw data and distilled knowledge of the teacher, improving their generalized performance (Menon et al., 2021). Therefore, recent works successfully demonstrated that a student with a larger or the same capacity can consistently exceed the teacher's performance to produce a more generalized model.

**Existing Distillation-based SSL.** Existing distillation-based SSL methods were mainly explored in the continuous domain (e.g., images, video) to remove the requirement of negative sampling for contrastive learning frameworks Grill et al. (2020a); Caron et al. (2021). They were also explored to reduce the size of the representation learning models (Abbasi Koohpayegani et al., 2020; Chen et al., 2020b). Many of these approaches combined KD with CL methods (Fang et al., 2021; Gao et al., 2022). SimCLR-v2 (Chen et al., 2020b) applied a larger teacher model, first trained using contrastive loss followed by supervised fine-tuning to distill a smaller model using the teacher. (Xu et al., 2020) incorporates auxiliary contrastive loss to obtain richer knowledge from the teacher network. A few other approaches also explored transferring the final embeddings of a self-supervised pre-trained teacher (Navaneet et al., 2022; Song et al., 2023).

## 2.3 Limitation of Existing Distillation-based SSL methods for graphs

Since a minor perturbation does not change the input semantics in the continuous domain, existing KD-based SSL methods did not focus on learning any semantic distance in the representation space for a pair of 'positive' samples. In contrast, we should design a novel representation learning framework for graphs that appropriately incorporate semantic differences due to minor discrete perturbations.

Our proposed TGCL first aims to obtain the teacher's distilled perception to calculate the semantic difference for any pairs, followed by formulating soft self-supervised losses to train the student. The notion of "distilled

perception" was previously explored in computer vision, typically to access the difference between two images or mitigating representations for semantically similar inputs. However, we introduce distilled distance for graphs to capture the semantic difference with subtle perturbations. Notably, such a notion is not necessary for computer vision applications as a minor perturbation does not change the semantics of an image.

Therefore, our proposed loss functions are specifically designed to accommodate the discrete nature of graphs and may not be appropriate for continuous domains (e.g., computer vision). To the best of our knowledge, such teacher-guided distilled distance has not been explored so far for representation learning (even in the vision domain) and is the key contribution of our paper.

## 3 Proposed Method

### 3.1 Preliminaries

***Graph Neural Network (GNN).*** Let $G = (V, E, X_V, X_E)$ be an undirected graph in the space of graphs $\mathcal{G}$, where $V, E, X_V, X_E$ denote the set of nodes, edges, node attributes, and edge attributes respectively. GNN encodes a graph $G \in \mathcal{G}$ to a $d$-dimensional embedding vector: $f : \mathcal{G} \to \mathbb{R}^d$. $f$ is often composed by stacking multiple message-passing layers. Let $h_v^{(l)}$ denote the representation of a node $v \in V$ having a neighborhood $N_v$ in the $l^{th}$ layer. $h_{vu}^{(l-1)}$ represents the attributes of edge $(v, u) \in E$ in the $(l-1)^{th}$ layer. Then, $h_v^{(l)}$ can be expressed as follows:

$$h_v^{(l)} = \phi_U^{(l-1)}\left(h_v^{(l-1)}, \underset{u \in N_v}{\oplus} \psi_M^{(l-1)}\big(h_v^{(l-1)}, h_u^{(l-1)}, h_{vu}^{(l-1)}\big)\right), \tag{1}$$

where $\phi_U^{(l-1)}, \psi_M^{(l-1)}$ are the update and the message function of $(l-1)^{th}$ layer respectively. $\oplus$ is a permutation invariant aggregator.

***Global Representations using Contrastive Learning.*** Contrastive learning (CL) aims to learn meaningful representations by attracting the positive pairs (*i.e.,* similar instances, such as two different perturbations of the same graph) while repelling negative pairs (*i.e.,* dissimilar instances, such as two different input graphs) in an unsupervised manner, as shown in Figure 2a. Formally, let $G_0$ denote the original graph and $G_p$ denote a perturbed version (*i.e.,* positive sample), and $\{G_{n_j}\}_j$ are other input graphs that are treated as negative sample. Then, the NT-Xent (normalized temperature-scaled cross-entropy) loss for CL objective for $G_0$ can be constructed as follows (Chen et al., 2020a; You et al., 2020):

$$\mathcal{L}_{CL} = -\log \frac{\exp\Big(sim\big(f(G_0), f(G_p)\big)/\tau\Big)}{\sum_{G \in G_p \cup \{G_{n_j}\}_j} \exp\Big(sim\big(f(G_0) \cdot f(G)\big)/\tau\Big)} \tag{2}$$

where $f$ is a GNN and $sim(\cdot, \cdot)$ is a similarity measure for embeddings with temperature-scaling $\tau$.

Minimization of Equation 2 brings positive pairs closer and pushes negative pairs further apart in the embedding space. However, unlike image augmentation schemes (*e.g.,* scaling, rotation, color jitter), graph augmentation schemes (*e.g.,* node/edge perturbations, subgraph sampling) may fail to preserve the graph semantics. For example, Figure 1(a) illustrates that removing one edge leads to two disconnected graphs, significantly changing the original semantics. Recently, D-SLA incorporates edit distances to introduce representational discrepancy even between graphs with minor perturbations (Kim et al., 2022). However, it only partly solves the issue.

### 3.2 Proposed TGCL Framework

This section presents our *Teacher-guided Graph Contrastive Learning (TGCL)* framework. Our proposed TGCL is fundamentally based on the following theoretical propositions. While these two propositions were proposed in two different literature of unsupervised contrastive learning and supervised distillation frameworks, they provide the perfect motivation to propose our teacher's distilled perception-guided contrastive learning framework for discrete domains of graphs.

**Proposition 1.** *Noise-contrastive estimation (NCE) to estimate the probability density function of a random variable is equivalent to training a binary classifier to distinguish between samples drawn from the true distribution and samples drawn from a noise distribution. The estimation of the true density function is derived from the learned binary classification function. (Gutmann & Hyvärinen, 2010)*

Proposition 1 indicates that a CL loss can be framed as a supervised classification loss where the network learns to separate *positive pairs* from *negative pairs* by artificially assigning "hard pseudo-labels" for each pair. Specifically, in Eq. 2, for $\mathcal{L}_{CL}$ formulation, we compute the probability for any arbitrary graph, $G' \in G_p \cup \{G_{n_j}\}_j$ constructing a positive pair with the original graph $G_0$ as a softmax function *i.e.*, $Pr(G_0, G') :=$

$$\frac{\exp\left(sim\left(f(G_0), f(G')\right)/\tau\right)}{\sum_{G \in G_p \cup \{G_{n_j}\}_j} \exp\left(sim\left(f(G_0) \cdot f(G)\right)/\tau\right)}.$$

Hence, by labeling the ground truth for a positive pair as $\boldsymbol{y}_{(G_0, G_p)} = 1$ and negative pairs as, $\boldsymbol{y}_{(G_0, G_{n_j})} = 0$ for all $j$, we can rewrite CL objective as: $\mathcal{L}_{CL} := -[\boldsymbol{y}_{(G_0, G_p)} \log Pr(G_0, G_p) + \sum_{G_{n_j}} \boldsymbol{y}_{(G_0, G_{n_j})} \log Pr(G_0, G_{n_j})]$, which is precisely the cross-entropy loss for classification.

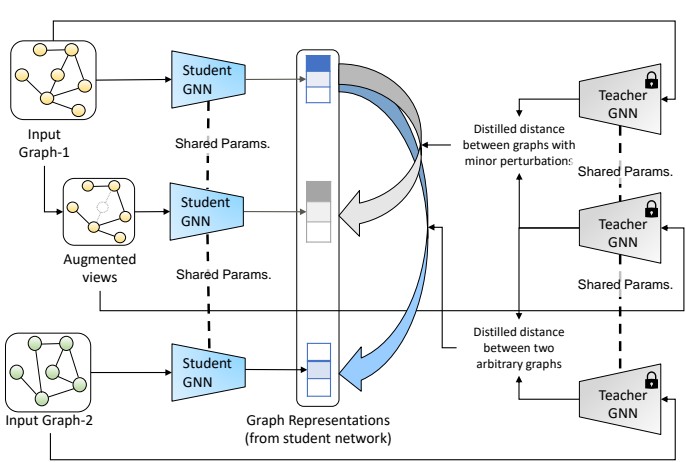

Figure 3: Block diagram of our proposed TGCL framework. We obtain the representations from a pre-trained teacher model and compute the distilled distance for each pair of inputs. These pairwise distances are employed to "soften" the loss functions to train the student.

Alternatively, we can construct a CL objective by directly distinguishing scores for *positive pairs* from *negative pairs* without computing explicit probabilities (You et al., 2020). Similarly, in D-SLA (Kim et al., 2022), both the graph discrimination loss and the edit-distance-based loss approach the task of distinguishing an anchor graph from a perturbed graph as a "hard" binary classification problem. Furthermore, the margin loss in D-SLA functions similarly to $\mathcal{L}_{cl}$, with a primary focus on separating positive and negative pairs.

**Proposition 2.** *In supervised learning, the variance of Bayesian-distilled risk, obtained using the soft labels, weighted by the likelihood from the Bayes teacher in the form of knowledge distillation (KD), remains lower than the variance of empirical risk, obtained using the 'hard' class-labels [Menon et al. (2021)]. That is,*

$$\mathbb{V}_{\overline{\mathcal{G}} \sim \mathcal{G}}[\hat{R}_*(f, \overline{\mathcal{G}})] \leq \mathbb{V}_{\overline{\mathcal{G}} \sim \mathcal{G}}[\hat{R}(f, \overline{\mathcal{G}})] \tag{3}$$

*where, $\mathbb{V}[\cdot]$ denotes the variance of a random variable. $|\cdot|$ is the cardinality. $\boldsymbol{p}^*(G_i)$ is the Bayes class-probability distribution, predicted using the Bayes teacher. $\hat{R}_*(f, \overline{\mathcal{G}}) = \frac{1}{|\overline{\mathcal{G}}|} \sum_{G \in \overline{G}} \boldsymbol{p}^*(G)^T \ell(f(G))$ is the Bayesian-distilled risk. $\hat{R}(f, \overline{\mathcal{G}}) = \frac{1}{|\overline{\mathcal{G}}|} \sum_{G \in \overline{G}}^{|\overline{\mathcal{G}}|} \boldsymbol{e}_G^T \ell(f(G))$ is the empirical risk where $\boldsymbol{e}_{G_i}$ are the hard 'class' labels for input graph $G_i$ $\ell$ is the empirical loss on $f$ (e.g., softmax cross-entropy). The equality holds iff $\forall G \in \mathcal{G}$, the loss values $\ell(f(G))$ are constant on the support of $\boldsymbol{p}^*(G)$.*

Proposition 2 suggests that irrespective of the *size/capacity* of a student model, it *statistically* produces a better generalization. In particular, it achieves a better generalization by using 'soft and distilled' labels for an existing CL method as the Bayes-distilled risk has lower variance compared to the naive un-distilled counterpart (Menon et al., 2021). Notably, this result does not depend on the capacity of the teacher or the student models.

Motivated by these results, we propose to obtain distilled perceptual distances, $\mathcal{D}_{dp}$ by comparing the deep features from a pre-trained teacher, followed by injecting them as "soft pseudo-labels" into the contrastive loss objective to learn the semantic differences between two arbitrary graphs in student's representation space.

---

**Algorithm 1:** PROPOSED TGCL FRAMEWORK

---

**Input:** Teacher model: $f_{teacher}$; Unlabelled Training set: $\mathcal{D}_{train}$

**Output:** Student Model: $f_s$

**1 for** *sampled mini-batch, $\mathcal{G}_B := \{G_i\}_i \in \mathcal{D}_{train}$* **do**

    /* Create multiple perturbations for each anchor graph.                                        */

**2**     $\{G_{p_{ij}}\}_j \xleftarrow[\text{variations}]{\text{perturbed}} G_i$

    /* Compute $\mathcal{D}_{dp}$ using $f_{teacher}$.                                                    */

**3**     Obtain $\mathcal{D}_{dp}(G_i, G_{p_{ij}}) \; \forall i, j$ between $G_i$ and $G_{p_{ij}}$

**4**     Obtain $\mathcal{D}_{dp}(G_i, G_{n_{ij}}) \; \forall i, j$ where $G_{n_{ij}} \in \mathcal{G}_B - G_i$

**5**     **if** *Framework == TGCL-GraphCL* **then**

**6**         $\min_{f_s} \sum_i \mathcal{L}_{TGCL-GraphCL}(G_i)$ (see Eq. 5)

**7**     **else if** *Framework == TGCL-DSLA* **then**

**8**         $\min_{f_s} \sum_i \mathcal{L}_{T-soft}(G_i; G_{p_{ij}}) + \lambda_1 \sum_i \mathcal{L}_{T-percept}(G_i; G_{p_{ij}}) + \lambda_2 \sum_i \mathcal{L}_{T-Margin}(G_i; G_{p_{ij}}, G_{n_{ij}})$

        (Eq. 11)

**Return:** Student Model, $f_s$

---

### 3.2.1 Distilled Perceptual Distance

Let $G_a$ and $G_b$ be two arbitrary graphs. Consider a representation learning model with $L$ message passing layers as the teacher. At each layer, $l$, we obtain the node-embedding $\{h_v^{(l)}\}_{v \in V}$ for a graph $G$ and apply a pooling operation (*e.g.*, max-pool) to obtain a fixed-length vector, $h_G^{(l)}$. We extract such fixed-length features from each layer and concatenate them, *i.e.*, $h_{G_a} = [\{h_{G_a}^{(l)}\}_l]$ and $h_{G_b} = [\{h_{G_b}^{(l)}\}_l]$ for $G_a$ and $G_b$ respectively. The *distilled perception distance (or distilled distance)* $\mathcal{D}_{dp}$ is then defined as the $L_2$ distance between these concatenated features, as:

$$\mathcal{D}_{dp}(G_a, G_b) = ||h_{G_a} - h_{G_b}||_2 \tag{4}$$

The *distilled distance* is similar to the "perceptual distance" in computer vision (Johnson et al., 2016). However, while we use distilled distance as a pseudo-label to inject semantic differences into the representation space, perceptual distance is employed to reduce gaps between semantically similar inputs in computer vision.

### 3.3 Proposed Loss Functions

The concept of teacher-guided loss with "softer" positive/negative pairs to train the student network can be introduced to any contrastive learning framework for graphs. To showcase the flexibility of our proposed TGCL framework, we present two versions of our framework: **(a)** TGCL-GraphCL & **(b)** TGCL-DSLA.

### 3.3.1 TGCL-GraphCL: TGCL using GraphCL Loss.

NT-Xent (normalized temperature-scaled cross-entropy) is a well-known loss function for contrastive learning models that have been widely explored for different domains, including graphs (Chen et al., 2020a; You et al., 2020).

**Loss Objective.** For GraphCL, the contrastive loss is obtained by applying the similarity function, sim (Eq. 2) using exponential of temperature scaled dot product of representations. In order to incorporate the "soft" distilled perception for a pair of graphs, when the distilled perceptual distance, $\mathcal{D}_{dp}(G_0, G_i)$ is smaller, we want the dot product of their representations, $f_s(G_0) \cdot f_s(G_i)$ to be higher. Therefore, we can balance their similarities by multiplying the teacher's distilled distance, $\mathcal{D}_{dp}(G_0, G_i)$ with the normalized dot product, $\frac{f_s(G_0) \cdot f_s(G_i)}{||f_s(G_0)|| \cdot ||f_s(G_i)||}$ for each pair of graphs. It leads to the following loss function:

$$\mathcal{L}_{TGCL-GraphCL} = \sum_{G_{p_i}} - \log \frac{\exp\left(\mathcal{D}_{dp}(G_0, G_{p_i}) \cdot \frac{1}{\tau} \cdot \frac{f_s(G_0) \cdot f_s(G_{p_i})}{||f_s(G_0)|| \cdot ||f_s(G_{p_i})||}\right)}{\sum_{G_{n_j}} \exp\left(\mathcal{D}_{dp}(G_0, G_{n_j}) \cdot \frac{1}{\tau} \cdot \frac{f_s(G_0) \cdot f_s(G_{n_j})}{||f_s(G_0)|| \cdot ||f_s(G_{n_j})||}\right)} \tag{5}$$

where, $f_s$ is the student network and $f_s(\cdot)$ is the representations obtained from $f_s$. $||\cdot||_2$ denotes the $L_2$ distance. $G_0$ is the anchor sample, $G_{p_i}$ is the $i^{th}$ perturbed sample. $G_{n_j}$ is $j^{th}$ negative sample for the anchor, $G_0$. Notably, while we use dot-product-based similarity in Eq. 5, any other similarity measures (*e.g.*, $L_2$-based) can also be used to design the loss function.

**Analysis.** In the numerator, for the positive pairs with a small distilled distance, $D_{dp}$, the student is forced to produce larger $\frac{f_s(G_0) \cdot f_s(G_{p_i})}{||f_s(G_0)|| \cdot ||f_s(G_{p_i})||}$ to minimize the overall loss. However, when $D_{dp}(G_0, G_{p_i})$ is large, the model does not receive the same incentive to maximize $\frac{f_s(G_0) \cdot f_s(G_{p_i})}{||f_s(G_0)|| \cdot ||f_s(G_{p_i})||}$ as before. The student would still maximize $\frac{f_s(G_0) \cdot f_s(G_{p_i})}{||f_s(G_0)|| \cdot ||f_s(G_{p_i})||}$ (at a smaller rate) to minimize the overall loss. Similarly, we can analyze the negative pairs in the denominator.

Note that our $\mathcal{L}_{TGCL-GraphCL}$ loss learns to discriminate the global representations of the whole graph without capturing local structural changes. Therefore, the TGCL-GraphCL framework is more appropriate for tasks related to global representations such as graph classification. In the following, we present the TGCL-DSLA framework for other tasks (such as link prediction) where the representations should also capture the local structural changes.

### 3.3.2  TGCL-DSLA: TGCL framework using D-SLA.

Next, we present TGCL-DSLA by introducing teacher-guided distilled perception distance for D-SLA. It consists of three components as follows:

**(a) Teacher-guided Soft Discrimination:** We first discriminate the perturbed graphs from the original anchor by introducing $\mathcal{L}_{T-Soft}$: It consists of two terms: The first one is a KD-based loss, $\mathcal{L}_{KD}$, while the second component is a weighted graph discrimination loss ($\mathcal{L}_{wGD}$). We first obtain the distilled distances: $[\mathcal{D}_{dp}(G_0, G_0), \{\mathcal{D}_{dp}(G_0, G_{p_i})\}_i]$ between the anchor, $G_0$, with itself and the $i^{th}$ perturbed variations, $G_{p_i}$. We obtain the similarities by taking reciprocals of the normalized distilled distance, followed by clipping to ensure numerical stability:

$$s_0 = \text{clip}\left(\frac{1}{\mathcal{D}_{dp}(G_0, G_0)}\right) \quad \text{and} \quad s_i = \text{clip}\left(\frac{1}{\mathcal{D}_{dp}(G_0, G_{p_i})}\right) \forall \, i > 0 \tag{6}$$

Next, we compute a probability distribution (soft labels) using the softmax-activation with temperature, $\tau$, *i.e.*, $\text{softmax}(s_0, s_1, \cdots; T = \tau)$. Similarly, we obtain a score for each graph and compute a probability distribution using temperature-scaled softmax: $\text{softmax}(\Psi \circ f_s(G_{p_i}); T = \tau)$. Now, we obtain the distillation loss, $\mathcal{L}_{KD}$ by minimizing the entropy between these probability distributions:

$$\mathcal{L}_{KD} := \tau^2 \mathcal{H}\left(\text{softmax}(s_0, s_1, \cdots; \tau), \text{softmax}(\Psi \circ f_s(G_{p_i}); \tau)\right) \tag{7}$$

where, $\mathcal{H}(y, \hat{y}) = \sum_y -y \log \hat{y}$ is the cross-entropy function. $\Psi$ is the scoring layer and $f_s$ is the student network. $\Psi \circ f_s$ is the composition of $\Psi$, and $f_s$. The representations obtained from $f_s$ are fed into the $\Psi$ layer to obtain the scores. Therefore, we incorporate the smoothened perception of the teacher in the score functions to learn the student's representations.

The second term, $\mathcal{L}_{wGD}$, is a set of *binary cross-entropy* functions with $G_0$ is labeled as 1 and $G_{p_i}$s' are labeled as 0 with the associated normalized soft-weights, $w_i = \frac{\mathcal{D}_{dp}(G_0, G_{p_i})}{\sum_i \mathcal{D}_{dp}(G_0, G_{p_i})}$:

$$\mathcal{L}_{wGD} = \mathcal{H}(1, \sigma(\Psi \circ f_s(G_0))) + \sum_i w_i \mathcal{H}(0, \sigma(\Psi \circ f_s(G_{p_i}))) \tag{8}$$

Therefore, $\mathcal{L}_{wGD}$ incorporates the teacher's soft label via $w_i$. Next, $\mathcal{L}_{T-soft}$ combines both components with a hyper-parameter $\alpha$:

$$\mathcal{L}_{T-soft} = \alpha \mathcal{L}_{KD} + (1 - \alpha)\mathcal{L}_{wGD} \tag{9}$$

**(b) Teacher-guided Perception Loss:** Next, we introduce a perception loss, $\mathcal{L}_{T-percept}$. It ensures that the embedding-level difference between original and perturbed graphs is proportional to the teacher's perspective of their corresponding distilled distances.

$$\mathcal{L}_{T-percept} = \sum_{i,j} \left( \frac{||f_s(G_{p_i}) - f_s(G_0)||_2}{\mathcal{D}_{dp}(G_{p_i}, G_0)} - \frac{||f_s(G_{p_j}) - f_s(G_0)||_2}{\mathcal{D}_{dp}(G_{p_j}, G_0)} \right)^2 \tag{10}$$

**(c) Teacher-guided Margin Loss:** The third component, $\mathcal{L}_{T-Margin}$ is a modified *margin loss* where the distilled distance acts as a regularizer, controlling the margin among the anchor, $G_0$, its perturbed variations, $G_{p_i}$ and the negative sample, $G_{n_j}$:

$$\mathcal{L}_{T-Margin} = \sum_{i,j} \max \left( 0, \ \beta_{ij} + ||f_s(G_{p_i}) - f_s(G_o)||_2 - ||f_s(G_{n_j}) - f_s(G_o)||_2 \right)$$

where, the margin $\beta_{ij} = \max \left( \beta, \ \mathcal{D}_{dp}(G_0, G_{n_j}) - \mathcal{D}_{dp}(G_0, G_{p_i}) \right)$ changes for each triplet based on teacher's perception.

***(d) Overall Loss:*** Finally, we obtain the overall loss by combining all three components as follows:

$$\mathcal{L}_{TGCL-DSLA} = \mathcal{L}_{T-soft} + \lambda_1 \mathcal{L}_{T-percept} + \lambda_2 \mathcal{L}_{T-Margin} \tag{11}$$

Where $\lambda_1$ and $\lambda_2$ are the hyper-parameters for training the

## 4 Experiments

In this section, we investigate the performance of TGCL for both TGCL-GraphCL and TGCL-DSLA frameworks on two diverse sets of experiments: (i) Graph Classification task in the chemical and biological domain and (ii) Link prediction on social network datasets. The graph classification task needs to capture the global structural representation of the graphs to improve the performance. In contrast, the link prediction task relies on the quality of capturing the local structural information. Therefore, it allows us to empirically compare our proposed TGCL-GraphCL and TGCL-DSLA frameworks and understand their effectiveness for different scenarios.

### 4.1 Main Results

#### 4.1.1 Graph Classification.

***Datasets.*** Following the prior works (You et al., 2021; Xu et al., 2021; Kim et al., 2022), we utilize *ZINC15* (Sterling & Irwin, 2015) to train the self-supervised representation learning models. Next, we finetune the models on eight different molecular benchmarks from MoleculeNet (Wu et al., 2018). We divide the datasets based on the constituting molecules' scaffold (molecular substructure). In table 1, we evaluate models' generalization ability on out-of-distribution test data samples (Wu et al., 2018).

We also present results from biological domains where the datasets are produced by the sampled ego networks from the PPI networks Zitnik et al. (2019). We use the same experimental setup as You et al. (2021) for predicting proteins' biological functions where we pre-train and fine-tune the model using the PPI network dataset Zitnik et al. (2019). In Table 4, we provide the dataset statistics.

***Evaluation Metric.*** We use the *Area Under Receiver Operating Characteristic curve (AUROC)* for benchmarking (Davis & Goadrich, 2006). AUROC quantifies the overall discriminative power of the classifier across all possible classification thresholds where higher values indicating better discrimination ability of the model. We report mean $\pm$ std with 5 independent runs.

***Performance Analysis.*** Table 1 compares with several different existing models along with *"no pretraining"* baselines. We can see that the *"no pretraining"* model achieves the least performance.

Table 1: AUROC score (%) comparison for molecular property prediction (i.e., graph classification) task. For our TGCL models, we indicate the corresponding teacher models within brackets. * - These models are specifically designed for molecular graphs that incorporate an *additional 50K 3D molecular graphs* of GEOM for training their self-supervised network.

| | Methods | BBBP | ClinTox | MUV | HIV | BACE | SIDER | Tox21 | ToxCast | *Avg* |
|---|---|---|---|---|---|---|---|---|---|---|
| | No Pretrain | 65.8 ± 4.5 | 58.0 ± 4.4 | 71.8 ± 2.5 | 75.3 ± 1.9 | 70.1 ± 5.4 | 57.3 ± 1.6 | 74.0 ± 0.8 | 63.4 ± 0.6 | 66.96 |
| Predictive | **Edgepred** (Hamilton et al., 2017) | 67.3 ± 2.4 | 64.1 ± 3.7 | 74.1 ± 2.1 | 76.3 ± 1.0 | 79.9 ± 0.9 | 60.4 ± 0.7 | 76.0 ± 0.6 | 64.1 ± 0.6 | 70.28 |
| | **AttrMasking** (Hu et al., 2020a) | 64.3 ± 2.8 | 71.8 ± 4.1 | 74.7 ± 1.4 | 77.2 ± 1.1 | 79.3 ± 1.6 | 61.0 ± 0.7 | 76.7 ± 0.4 | 64.2 ± 0.5 | 71.15 |
| | **ContextPred** (Hu et al., 2020a) | 68.0 ± 2.0 | 65.9 ± 3.8 | 75.8 ± 1.7 | 77.3 ± 1.0 | 79.6 ± 1.2 | 60.9 ± 0.6 | 75.7 ± 0.7 | 63.9 ± 0.6 | 70.89 |
| | **GraphMAE** Hou et al. (2022) | 70.1 ± 0.6 | 80.8 ± 1.2 | 74.5 ± 2.3 | 77.0 ± 0.4 | 81.4 ± 0.9 | 59.0 ± 0.7 | 74.4 ± 0.5 | 63.9 ± 0.4 | 72.64 |
| Contrastive | **Infomax** (Veličković et al., 2019) | 68.8 ± 0.8 | 69.9 ± 3.0 | 75.3 ± 2.5 | 76.0 ± 0.7 | 75.9 ± 1.6 | 58.4 ± 0.8 | 75.3 ± 0.5 | 62.7 ± 0.4 | 70.29 |
| | **GraphCL** (You et al., 2020) | 69.7 ± 0.7 | 76.0 ± 2.7 | 69.8 ± 2.7 | 78.5 ± 1.2 | 75.4 ± 1.4 | 60.5 ± 0.9 | 73.9 ± 0.7 | 62.4 ± 0.6 | 70.78 |
| | **JOAO** (You et al., 2021) | 70.2 ± 1.0 | 81.3 ± 2.5 | 71.7 ± 1.4 | 76.7 ± 1.2 | 77.3 ± 0.5 | 60.0 ± 0.8 | 75.0 ± 0.3 | 62.9 ± 0.5 | 71.89 |
| | **JOAOv2** (You et al., 2021) | 71.4 ± 0.9 | 81.0 ± 1.6 | 73.7 ± 1.0 | 77.7 ± 1.2 | 75.5 ± 1.3 | 60.5 ± 0.7 | 74.3 ± 0.6 | 63.2 ± 0.5 | 72.16 |
| | **GraphLoG** (Xu et al., 2021) | 72.5 ± 0.8 | 76.7 ± 3.3 | 76.0 ± 1.1 | 77.8 ± 0.8 | 83.5 ± 1.2 | 61.2 ± 1.1 | 75.7 ± 0.5 | 63.5 ± 0.7 | 73.36 |
| | **BGRL** (Thakoor et al., 2022) | 66.7 ± 1.7 | 64.7 ± 6.5 | 69.4 ± 2.7 | 75.5 ± 1.9 | 71.3 ± 5.5 | 60.4 ± 1.4 | 74.8 ± 0.7 | 63.2 ± 0.8 | 68.25 |
| | **SimGCL** (Yu et al., 2022) | 67.4 ± 1.2 | 55.7 ± 4.7 | 71.2 ± 1.8 | 75.0 ± 0.9 | 74.1 ± 2.7 | 57.4 ± 1.7 | 74.4 ± 0.5 | 62.3 ± 0.4 | 67.19 |
| | **SimGRACE** (Xia et al., 2022) | 71.3 ± 0.9 | 64.2 ± 4.5 | 71.2 ± 3.4 | 74.5 ± 1.1 | 73.8 ± 1.4 | 60.59 ± 0.9 | 74.2 ± 0.6 | 63.4 ± 0.5 | 69.13 |
| | **D-SLA** (Kim et al., 2022) | 72.6 ± 0.8 | 80.2 ± 1.5 | 76.6 ± 0.9 | 78.6 ± 0.4 | 83.8 ± 1.0 | 60.2 ± 1.1 | 76.8 ± 0.5 | 64.2 ± 0.5 | 74.13 |
| + Additional Data (*) | **3D-InfoMax** Stärk et al. (2022) | 67.9 ± 1.2 | 89.7 ± 0.5 | 76.7 ± 0.6 | 73.4 ± 1.2 | 79.9 ± 0.9 | 59.6 ± 0.7 | 75.3 ± 0.3 | 64.6 ± 0.4 | 73.4 |
| | **GraphMVP-G** Liu et al. (2022) | 70.1 ± 0.7 | 89.4 ± 1.5 | 77.7 ± 1.6 | 75.3 ± 0.8 | 80.2 ± 1.5 | 61.0 ± 0.5 | 75.3 ± 0.9 | 64.2 ± 0.9 | 74.1 |
| | **FragCL** Kim et al. (2023) | 71.4 ± 0.4 | **95.2 ± 1.0** | 77.6 ± 1.0 | 76.3 ± 0.4 | 82.3 ± 1.6 | 61.0 ± 0.6 | 75.2 ± 0.7 | **65.1 ± 0.8** | 75.5 |
| Ours | **TGCL-GraphCL (w/ GraphLoG)** | **74.9 ± 0.9** | 85.3 ± 2.2 | 78.9 ± 1.0 | **79.1 ± 0.5** | 83.7 ± 1.4 | 63.6 ± 0.6 | 76.7 ± 0.4 | 64.1 ± 0.4 | **75.79** |
| | **TGCL-GraphCL (w/ D-SLA)** | 74.0 ± 0.4 | 82.8 ± 2.2 | 77.0 ± 0.9 | 77.9 ± 0.3 | 84.3 ± 1.0 | **64.2 ± 0.3** | 76.6 ± 0.1 | 64.7 ± 0.4 | 75.19 |
| | **TGCL-DSLA (w/ GraphLoG)** | 74.8 ± 0.3 | 80.6 ± 0.5 | 77.4 ± 0.1 | 78.6 ± 0.2 | 83.0 ± 1.1 | 61.4 ± 0.4 | 76.1 ± 0.1 | 64.0 ± 0.3 | 74.49 |
| | **TGCL-DSLA (w/ D-SLA)** | 73.5 ± 0.9 | **84.9 ± 1.3** | **79.4 ± 0.9** | 78.8 ± 0.5 | **85.2 ± 0.4** | 61.2 ± 1.0 | **76.9 ± 0.1** | 64.9 ± 0.2 | 75.60 |

**(a) Predictive vs. Contrastive Models.** While predictive pretraining improves upon the no pertaining model, their performance remains worse than the CL models. This is because predictive methods primarily focus on the local structure, while molecular properties depend on the global structure. In contrast, CL methods focus on the global structure by contrasting between original and perturbed graphs to achieve better performance. While a few augmentation-free CL (Yu et al., 2022; Xia et al., 2022) methods are proposed, their performance remains significantly lower than the state-of-the-art. GraphLoG achieves higher AUROC scores by exploring both global semantics and local substructures. However, D-SLA achieves the best performance among the existing models by exploring the local discrete graph structures.

**(b) Performace of Proposed TGCL.** For our TGCL, we report the results by using GraphLoG and D-SLA as teacher modules for both TGCL-GraphCL and TGCL-DSLA models, demonstrating the generalizability of our framework. We observe that our proposed method consistently boosts the performance of teachers irrespective of the teacher's training methodology for both TGCL models. Furthermore, we also outperformed the existing molecular graph-specific representation learning models that incorporate additional 3D molecular graphs of GEOM for training their representation models.

Table 2: AUROC score(%) for PPI dataset. For our TGCL models, we indicate the corresponding teacher models within brackets.

| Methods | AUROC |
|---|---|
| **DSLA** | 71.56 ± 0.46 |
| **GraphLog** | 66.92 ± 1.58 |
| **TGCL-GraphCL (w/ GraphLoG)** | 71.96 ± 0.77 |
| **TGCL-DSLA (w/ D-SLA)** | 71.63 ± 0.96 |

Next, we observe that TGCL-GraphCL (w/ GraphLoG) performs best. Also, TGCL-DSLA (w/ D-SLA) achieves comparable performance as TGCL-GraphCL (w/ GraphLoG). In particular, we do not observe an additional advantage of using more graph-specific D-SLA-based loss functions while learning global graph-level representations for molecular property prediction tasks.

***Performance on PPI network.*** In table 2, we compare with GraphLoG and D-SLA and the corresponding student i.e., TGCL-GraphCL (w/ GraphLoG) and TGCL-DSLA (w/ D-SLA). The results show that the student models consistently outperform their corresponding teachers. However, the observed improvements are not substantial, and the teacher models have already approached their performance saturation point Furlanello et al. (2018). This phenomenon, known as knowledge saturation, is further explored in Appendix A.3.2.

***Visualizing the learned latent representation space.*** In Figure 4, we visualize the learned latent representation space of D-SLA and our TGCL-DSLA (w/ DSLA) utilizing t-SNE (Van der Maaten & Hinton, 2008). We observe that TGCL-DSLA segregates the positive and negative samples more successfully than

DSLA. Therefore, it indicates that our TGCL-DSLA (w/ DSLA) produces a better representation to easily predict the molecular properties.

### 4.1.2 Link Prediction.

***Datasets.*** We consider COLLAB, IMDB-Binary, and IMDB-Multi from TU Benchmarks (Morris et al., 2020). We separate the dataset into four parts: pre-training, training, validation, and test sets in the ratio of 5:1:1:3, as in Kim et al. (2022). Additional details are provided in Table 5 (Appendix).

***Evaluation Metric.*** Table 3 presents comparative results with several existing models and "No Pretrain" baselines. We compare average precision (as in (Kim et al., 2022)) where a higher value indicates better performance. We report mean±std for 5 independent runs.

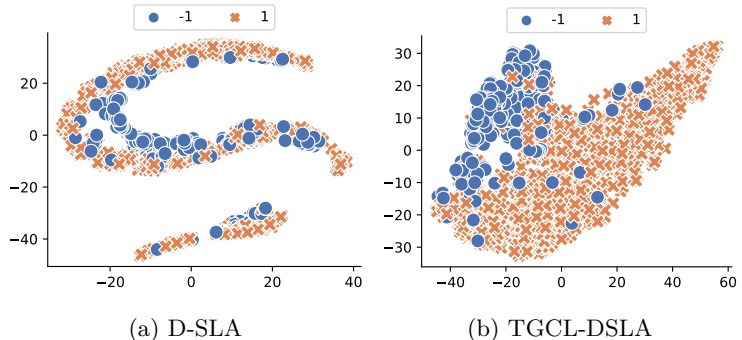

(a) D-SLA       (b) TGCL-DSLA

Figure 4: t-SNE visualization of representations for BBBP dataset using (a) D-SLA teacher and (b) TGCL-DSLA.

***Performance Analysis.***

**(a) Predictive vs. Contrastive Models** Unlike graph classification tasks, local context plays a crucial role in link prediction. Therefore, the predictive models typically outperformed most of the CL methods. Among the existing CL methods, GraphLog performs similarly to ContextPred as it focuses on both local and global structures. D-SLA performs better by capturing local structures using edit-distance-based discriminations that standard CL models fail to distinguish.

**(b) Performace of Proposed TGCL.** In comparison, our proposed distilled distance from the teacher network integrates a regularized approach to both local and global semantics. Local semantics are captured from the initial latent features, while global semantics are derived from deeper global features. Therefore, we can surpass existing local and global representation learning-based models by visible margins for all three datasets. Interestingly, our TGCL-DSLA (w/GraphLog) performs better than TGCL-DLSA (w/D-SLA) even though D-SLA outperformed

Table 3: Average precision score(%) comparison on link prediction task on social networks. For our TGCL models, we indicate the corresponding teacher models within brackets.

| Methods | COLLAB | IMDB-Binary | IMDB-Multi | Avg. |
|---|---|---|---|---|
| **No Pretrain** | 80.01 ± 1.14 | 68.72 ± 2.58 | 64.93 ± 1.92 | 71.22 |
| **AttrMasking** (Hu et al., 2020a) | 81.43 ± 0.80 | 70.62 ± 3.68 | 63.37 ± 2.15 | 71.81 |
| **ContextPred** (Hu et al., 2020a) | 83.96 ± 0.75 | 70.47 ± 2.24 | 66.09 ± 2.74 | 73.51 |
| **Infomax** (Veličković et al., 2019) | 80.83 ± 0.62 | 67.25 ± 1.87 | 64.98 ± 2.47 | 71.02 |
| **GraphCL** (You et al., 2020) | 76.04 ± 1.04 | 63.71 ± 2.98 | 62.40 ± 3.04 | 67.38 |
| **JOAO** (You et al., 2021) | 76.57 ± 1.54 | 65.37 ± 3.23 | 62.76 ± 1.52 | 68.23 |
| **GraphLoG** (Xu et al., 2021) | 82.95 ± 0.98 | 69.71 ± 3.18 | 64.88 ± 1.87 | 72.51 |
| **BGRL** (Thakoor et al., 2022) | 76.79 ± 1.13 | 67.97 ± 4.14 | 63.71 ± 2.09 | 69.49 |
| **SimGCL** (Yu et al., 2022) | 77.46 ± 0.86 | 64.91 ± 2.60 | 63.78 ± 2.28 | 68.72 |
| **SimGRACE** (Xia et al., 2022) | 74.51 ± 1.54 | 64.49 ± 2.79 | 62.81 ± 2.32 | 67.27 |
| **D-SLA** (Kim et al., 2022) | 86.21 ± 0.38 | 78.54 ± 2.79 | 69.45 ± 2.29 | 78.07 |
| **TGCL-GraphCL (w/ GraphLoG)** | 87.23 ± 0.14 | 75.09 ± 1.88 | 67.11 ± 3.73 | 76.48 |
| **TGCL-GraphCL (w/ D-SLA)** | 87.51 ± 1.24 | 77.95 ± 3.89 | 67.88 ± 2.20 | 77.78 |
| **TGCL-DSLA (w/ GraphLoG)** | **91.09 ± 0.33** | **83.15 ± 0.89** | **74.11 ± 1.44** | **82.78** |
| **TGCL-DSLA (w/ D-SLA)** | 87.51 ± 0.59 | 80.03 ± 4.13 | 70.97 ± 2.42 | 79.50 |

GraphLog. Therefore, a better teacher does not necessarily produce better distillation for the student, as previously observed and analyzed in supervised learning Menon et al. (2021); Kaplun et al. (2022); Zong et al. (2023).

### 4.1.3 TGCL-GraphCL vs. TGCL-DSLA: Choosing the correct framework for downstream tasks.

As we can see for the graph classification task (Table 1), our TGCL-GraphCL framework achieves better performance while for link prediction tasks (Table 3), TGCL-DSLA produces better results. Therefore, these empirical results indicate that TGCL-GraphCL produces better global representations for graphs, allowing us to easily distinguish two arbitrary graphs in inductive settings. Hence, this framework is better suited

for graph classification tasks. In contrast, TGCL-DSLA also effective in capturing the local structural information by explicitly learning to distinguish the anchor and augmented samples (using $\mathcal{L}_{T-soft}$ [Eq. 7] and $\mathcal{L}_{T-percept}$ [Eq. 10]). Hence, it leads to better performance for the link prediction task in transductive settings. In summary, we should choose TGCL-GraphCL for inductive settings and TGCL-DSLA for transductive settings.

## 5  Conclusion & Discussion

We utilize Knowledge distillation (KD) for graph representation learning, where a self-supervised pre-trained teacher is used to guide a student model to produce more generalized representations. Extensive experimentation demonstrates the effectiveness of the proposed method in improving the performance of graph classification and link prediction tasks. However, there are still many open challenges in graph representation learning, such as the efficient handling of large-scale graphs, the ability to handle heterogeneity and multimodality, and the development of robust methods for noisy or incomplete data. Probing these challenges further and developing new graph representation learning techniques are in the scope of future research.

***Discussion. Potential Advantages and Limitations of KD for CL***
***Advantages.***  KD typically enhances a CL framework to produce improved feature representations by transferring insights from a well-trained teacher, improving the downstream performance (Hinton et al., 2015; Furlanello et al., 2018; Yim et al., 2017). Further, by utilizing the teacher's knowledge, we can speed up the training time efficiency and reduce the sample requirement to learn the student model (Tian et al., 2019). A well-trained teacher can also improve the student model's robustness, leading to even better representations compared to the teacher. Finally, regularizing the student model with softer probability scores reduces the variance of Bayes-distilled risk, therefore, making the model more generalizable and less prone to overfitting (Menon et al., 2021).

***Limitations.***      The teacher-student framework for knowledge distillation may face several potential challenges – (a) KD with CL adds both computation overhead and training complexity (Hinton et al., 2015; Tian et al., 2019). (a) If the *teacher model is flawed*, the student may inherit its errors, leading to suboptimal performance (Hinton et al., 2015). (b) *Compatibility issues* between teacher and student models and *training instability* can hinder effective knowledge transfer. It requires intricate tuning of hyperparameters and designing the loss functions. (Mirzadeh et al., 2020). (c) Further, the *generalization* may be restricted due to limited knowledge of the teacher with respect to the downstream tasks Yim et al. (2017). (d) Finally, there is a risk of *knowledge saturation*, where the benefits of additional knowledge transfer may diminish after a certain saturation point (Furlanello et al., 2018).

Appendix A.3 and A.4 provide additional ablation studies to empirically investigate these challenges.

***Broader Impact.***      KD can significantly impact graph representations, with broader implications for various fields, including bioinformatics, drug discovery, social network analysis, recommendation systems, etc. A few potential impacts of our work are as follows: (a) Improves the efficiency and scalability of graph representation learning by enabling 'soft' knowledge transfer from a pre-trained teacher model to a smaller, more efficient student network. (b) Improves the generalization performance of graph representation learning by leveraging the 'dark knowledge' encoded in a pre-trained teacher model's representations.

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

# A  Experimental Setup

## A.1  Molecular graph classification

This section presents the implementation details and dataset descriptions of our experiments on molecular graph classification and link prediction tasks. For all experiments, we use PyTorch (Paszke et al., 2019) and PyTorch Geometric libraries (Fey & Lenssen, 2019) with a single NVIDIA A30 Tensor Core GPU for all of our experiments.

**Datasets.**  For our first experiments on molecular graph classification, we use the ZINC dataset (Sterling & Irwin, 2015), a large corpus of  2 million unlabelled molecules for pretraining the teacher and student network. For the downstream tasks, we experimented with 8 labeled molecular datasets from MolecularNet (Wu et al., 2018). The molecule classes are determined using the biophysical and physiological properties.

We also present results from biological domains where the datasets are produced by the sampled ego networks from the PPI networks Zitnik et al. (2019). We use the same experimental setup as You et al. (2021) for predicting proteins' biological functions where we pre-train and fine-tune the model using the PPI network dataset Zitnik et al. (2019). In Table 4, we provide the statistics of these datasets.

Table 4: Descriptions of Molecular and PPI network datasets.

| Chemical Datasets | #Graphs | Avg. Nodes | Avg. Edges | Tasks |
|---|---|---|---|---|
| ZINC15 | 2,000,000 | 26.62 | 28.86 | - |
| BBBP | 2,039 | 24.06 | 25.95 | 1 |
| ClinTox | 1,478 | 26.16 | 27.88 | 2 |
| MUV | 93,087 | 24.23 | 26.28 | 17 |
| HIV | 41,127 | 25.51 | 27.47 | 1 |
| BACE | 1,513 | 34.09 | 36.86 | 1 |
| SIDER | 1,427 | 33.64 | 35.36 | 27 |
| Tox21 | 7,831 | 18.57 | 19.29 | 12 |
| ToxCast | 8,575 | 18.78 | 19.26 | 617 |
| **Biological Datasets** | **#Graphs** | **Avg. Nodes** | **Avg. Edges** | **Tasks** |
| PPI (Pre-training) | 306,925 | 39.83 | 364.82 | - |
| PPI (Finetune) | 88,000 | 49.35 | 445.39 | 40 |

**Implementation details.**  For our proposed framework, we use the same network architecture for both the teacher and the student model. In particular, we use Graph Isomorphism Networks (GINs) (Xu et al., 2019) as applied in the previous works Hu et al. (2020a); Xu et al. (2021); Kim et al. (2022). These networks consist of 5 layers with 300 dimensional embeddings for nodes and edges along with average pooling strategies for obtaining the graph representations. To obtain distilled perception distance from the teacher network, we use global average pooling to extract the fixed-length features from each layer.

We use the official D-SLA codes[1] provided by Kim et al. (2022) as the backbone for our experiments and apply the same perturbation strategies as used in (Kim et al., 2022). In particular, their perturbation strategy aims to minimize the risk of creating unreasonable cycles, reducing the chance of significant change in the chemical properties. For our experiments, we use three perturbations for each input sample.

We report results using two different teacher modules, trained using existing GraphLog (Xu et al., 2021) and D-SLA (Kim et al., 2022) while training the following student network using the loss functions as proposed in Section 3.3. We divide the perceptual distances by 4 and 1 as we use GraphLog (Xu et al., 2021) and D-SLA (Kim et al., 2022) as the teacher, respectively. For TGCL-GraphCL, we use $\tau = 10$ in Equation 5. For TGCL-DSLA, we use $\lambda_1$ and $\lambda_2$ to 1.0 and 0.5 respectively for the student model. For $\mathcal{L}_{T-soft}$ loss, we set the temperature, $\tau = 10$ (Equation 7) and $\alpha = 0.95$ (Equation 9). For $\mathcal{L}_{T-margin}$, we set $\beta = 5$. Both teacher and student models are trained using batch-size of 256 and for 25 epochs with learning rate $1e - 3$ and Adam optimizer (Kingma & Ba, 2014).

---

[1]https://github.com/dongkikim95/d-sla

## A.2 Link Prediction

**Datasets.** For this task, we select three datasets i.e., COLLAB, IMDB-Binary, and IMDB-Multi from the TU dataset benchmark Morris et al. (2020).

Table 5: Statistics of social network datasets for link prediction.

| Datasets | #Graphs | Avg. Nodes | Avg. Edges |
|---|---|---|---|
| **COLLAB** | 4320 | 76.12 | 2331.4 |
| **IMDB-B** | 2039 | 20.13 | 85.5 |
| **IMDB-M** | 1478 | 16.64 | 77.9 |

COLLAB is a dataset of scientific collaboration networks. It contains $4,320$ graphs where the researcher and their collaborators are nodes and an edge indicates collaboration between two researchers. A researcher's ego network has three possible labels: *High Energy Physics*, *Condensed Matter Physics*, and *Astro Physics*, representing the field of the researchers.

IMDB-Binary is a movie collaboration dataset. It consists of the ego networks of actors/actresses from the movies in IMDB. It consists of $2,039$ graphs. For each graph, the nodes are actors/actresses, with an edge between them if they appear in the same movie. These graphs are derived from the Action and Romance genres.

IMDB-Multi is a relational dataset that consists of a network of actors or actresses, played in movies in IMDB. It contains 1,478 graphs. As before, a node represents an actor or actress, and an edge connects two nodes when they appear in the same movie. The edges are collected from three different genres: Comedy, Romance, and Sci-Fi.

**Implementation details.** For our experiments, we use Graph Convolutional Network (GCN) Kipf & Welling (2017) for both teacher and student models. These networks consist of three layers with 300 dimensions for embeddings. As before, we use the same perturbation strategy as applied in Kim et al. (2022). We have also experimented with two different teachers, *i.e.*, D-SLA Kim et al. (2022) and GraphLog Xu et al. (2021). We use a batch size of 32 and a learning rate of 0.001 for training the student representation learning models. For TGCL-GraphCL, we set $\tau = 10$ (Eq. 5). For TGCL-DSLA, we use $\lambda_1$ and $\lambda_2$ to 0.7 and 0.0, respectively. For $\mathcal{L}_{T-soft}$ loss, we select the temperature, $\tau$ from three different values i.e., $\{5, 10, 20\}$ (Equation 7) and set $\alpha = 0.95$ (Eq. 9).

## A.3 Ablation Study on Teacher-Student Framework for Graph COntrastive Learning

In this section, we evaluate the performance of TGCL models with various teacher models and model capacities. We also analyze the trade-off between computational overhead and convergence by applying undertrained teacher models vs. a well-trained teacher model.

### A.3.1 Student Network with reduced Capacity

The original goal of Knowledge Distillation (KD) (Hinton et al., 2015) was to transfer knowledge from a complex 'teacher' model to a simpler 'student' model, achieving comparable performance with reduced computational cost. However, subsequent studies demonstrate that using a complex student model, similar to the teacher, can also enhance model robustness, efficiency, and generalization across various applications (Furlanello et al., 2018; Tian et al., 2019). Although our main experiments utilize identical architectures for both teacher and student models, this section explores the impact of employing a smaller student network.

Table 6 demonstrates the performance of a student TGCL-DSLA model on the downstream molecular property prediction task. We can see that, with the same capacity (i.e., 5 layers of GNN) as the teacher module of D-SLA, our proposed student network consistently outperformed the teacher. As we decrease the capacity of our student network by reducing the number of layers, the overall performance reduces. However, we observe that even with 3 layers of GNN, our student module outperforms the teacher D-SLA model. Therefore, these results demonstrate that our proposed TGCL framework can compress the student

representation network by enabling smoothened knowledge transfer from a pre-trained teacher to the student representation learning model.

Table 6: Impact of the capacity of the student TGCL-DSLA models (mean ± std) for graph classification. "Full-capacity" denotes the same capacity as the teacher.

| | BBBP | ClinTox | MUV | HIV | BACE | SIDER | Tox21 | ToxCast | Avg |
|---|---|---|---|---|---|---|---|---|---|
| **D-SLA** (Kim et al., 2022) | 72.6 ± 0.8 | 80.2 ± 1.5 | 76.6 ± 0.9 | 78.6 ± 0.4 | 83.8 ± 1.0 | 60.2 ± 1.1 | 76.8 ± 0.5 | 64.2 ± 0.5 | 74.13 |
| **w/ full-capacity** | 73.5 ± 0.9 | **84.9 ± 1.3** | **79.4 ± 0.9** | **78.8 ± 0.5** | **85.2 ± 0.4** | **61.2 ± 1.0** | **76.9 ± 0.1** | **64.9 ± 0.2** | **75.60** |
| **w/ 3-layer Student** | **74.6 ± 0.4** | 84.6 ± 1.4 | 76.4 ± 1.0 | 77.9 ± 0.1 | 82.7 ± 1.1 | 61.0 ± 0.3 | 75.0 ± 0.1 | 63.6 ± 0.4 | 74.48 |
| **w/ 2-layer Student** | 72.6 ± 0.5 | 81.4 ± 0.4 | 77.3 ± 1.5 | 77.6 ± 0.2 | 80.6 ± 0.4 | 60.8 ± 0.4 | 74.7 ± 0.4 | 63.0 ± 0.1 | 73.50 |

### A.3.2 TGCL with multi-level teachers & Knowledge Saturation

Proposition 2 suggests that irrespective of the size/capacity of a student model, it statistically produces a better generalization. Hence, it raises a natural question: *can we further improve the performance by iteratively using the student models as the teacher to train another follow-up student network?* In Table 7, we investigate this by training a 2-level iterative teacher model for our TGCL-DSLA framework. In other words, we use the *TGCL-DSLA (w/ D-SLA)* student model to train another 2nd-level student, denoted as TGCL$^2$-DSLA (w/ D-SLA).

We can see that the performance of TGCL$^2$-DSLA (w/ D-SLA) saturates and does not improve the overall performance than the original TGCL-DSLA (w/ D-SLA). These results indicate that the TGCL-DSLA already receives sufficient probability calibrations from the first-level teacher model. Hence, their performance improvement converges after the first-level teacher. This phenomenon is known as "knowledge saturation" in the KD literature Furlanello et al. (2018).

Table 7: Performance comparison of TGCL models with two-level teachers.

| | BBBP | ClinTox | MUV | HIV | BACE | SIDER | Tox21 | ToxCast | Avg. |
|---|---|---|---|---|---|---|---|---|---|
| **DSLA** | 72.6 ± 0.8 | 80.2 ± 1.5 | 76.6 ± 0.9 | 78.6 ± 0.4 | 83.8 ± 1.0 | 60.2 ± 1.1 | 76.8 ± 0.5 | 64.2 ± 0.5 | 74.1 |
| **TGCL-DSLA (w/ D-SLA)** | 73.5 ± 0.9 | **84.9 ± 0.9** | **79.4 ± 0.9** | **78.8 ± 0.5** | **85.2 ± 0.4** | 61.2 ± 1.0 | **76.9 ± 0.1** | **64.9 ± 0.2** | **75.6** |
| **TGCL$^2$-DSLA (w/ D-SLA)** | **74.15 ±0.9** | 84.65 ±2.0 | 76.43 ±1.3 | 78.67 ±0.4 | 84.07 ± 0.8 | **62.73 ± 0.8** | 76.15 ± 0.4 | 64.32 ± 0.4 | 75.2 |

### A.3.3 Choice of teacher models: Computational overhead and convergence

KD enhances a CL framework by improving feature representations but introduces additional computational overhead and training complexity. In Table 8, we examine the performance of the TGCL student model using an undertrained teacher model with early stopping.

We observe that the student model consistently outperforms the teacher models, even when the teacher models are undertrained. For instance, our TGCL-GraphCL model with D-SLA (trained for 20 epochs) achieves an average AUROC score of 74.4%, surpassing the performance of D-SLA model trained for 100 epochs. These results suggest that the TGCL student model can achieve comparable performance without requiring a fully trained teacher model. Consequently, it is possible to enhance downstream performance with only a modest increase in computational overhead.

Table 8: Performance comparison of TGCL students using an undertrained teacher model with early stopping.

| | BBBP | ClinTox | MUV | HIV | BACE | SIDER | Tox21 | ToxCast | Avg. |
|---|---|---|---|---|---|---|---|---|---|
| **DSLA [100 epochs]** | 72.6 ± 0.8 | 80.2 ± 1.5 | 76.6 ± 0.9 | 78.6 ± 0.4 | 83.8 ± 1.0 | 60.2 ± 1.1 | 76.8 ± 0.5 | 64.2 ± 0.5 | 74.1 |
| **TGCL-GraphCL (w/ D-SLA [20 epochs])** | 73.3 ± 0.1 | 82.7 ± 0.8 | 76.0 ± 1.1 | 78.5 ± 0.8 | 82.8 ± 0.8 | 62.4 ± 0.2 | 75.1 ± 0.2 | 64.2 ± 0.4 | 74.4 |
| **TGCL-GraphCL (w/ D-SLA [60 epochs])** | 73.4 ±0.6 | 84.3 ±1.3 | 75.6 ±0.8 | **79.3 ±0.6** | **84.9 ± 0.6** | 62.0 ± 0.6 | 75.4 ± 0.2 | 63.7 ± 0.2 | 74.8 |
| **TGCL-GraphCL (w/ D-SLA [100 epochs])** | **74.0 ±0.4** | **82.8 ±2.2** | **77.0 ±0.9** | 77.9 ±0.3 | 84.3 ±1.0 | **64.2 ± 0.3** | **76.6 ± 0.1** | **64.7 ± 0.4** | 75.2 |

### A.4 Ablation Studies on hyper-parameters and loss components

In this section, we investigate the effect of different hyperparameters and contributions of different loss components for our TGCL framework and the sensitivity of their hyperparameters.

#### A.4.1 Impact of different loss components.

In Table 9, we first demonstrate the impact of different loss components. The first three rows demonstrate the performance of individual loss components. We observe that $\mathcal{L}_{soft}$ is the most essential component, providing the maximum performance boost for the downstream molecular prediction tasks. The other two loss components, i.e. $\mathcal{L}_{T-percept}$ and $\mathcal{L}_{T-margin}$ act as regularizer. While, individually, they do not perform well, incorporating them with $\mathcal{L}_{soft}$ in $\mathcal{L}_{overall}$, we observe a significant boost in the overall performance.

Table 9: Impact of individual loss components. (at $\alpha = 0.95$, $\tau = 10$)

| $\mathcal{L}_{T-soft}$ | $\mathcal{L}_{T-percept}$ | $\mathcal{L}_{T-margin}$ | BBBP | ClinTox | MUV | HIV | BACE | SIDER | Tox21 | ToxCast | $Avg$ |
|---|---|---|---|---|---|---|---|---|---|---|---|
| ✓ | ✗ | ✗ | 72.9 ± 1.4 | 79.8 ± 1.2 | 79.1 ± 0.7 | 77.7 ± 0.6 | 81.9 ± 0.3 | 62.1 ± 0.7 | **76.9 ± 0.2** | 64.1 ± 0.3 | 74.31 |
| ✗ | ✓ | ✗ | 71.6 ± 0.8 | 74.5 ± 0.7 | 76.6 ± 1.3 | 78.5 ± 1.1 | 81.7 ± 0.9 | 61.7 ± 0.6 | 75.7 ± 0.6 | 62.9 ± 0.3 | 72.90 |
| ✗ | ✗ | ✓ | 72.7 ± 0.6 | 77.9 ± 2.0 | 74.1 ± 0.9 | 76.6 ± 0.4 | 82.9 ± 0.6 | **62.8 ± 0.5** | 74.2 ± 0.1 | 61.9 ± 0.8 | 72.89 |
| ✗ | ✓ | ✓ | **73.6 ± 0.5** | 81.2 ± 1.1 | 75.7 ± 0.4 | 77.3 ± 1.4 | 83.2 ± 0.3 | **62.8 ± 0.6** | 75.2 ± 0.2 | 63.3 ± 0.5 | 74.04 |
| ✓ | ✓ | ✗ | 72.8 ± 0.1 | 81.6 ± 0.5 | 79.2 ± 0.5 | **78.8 ± 0.9** | 81.4 ± 1.2 | 59.7 ± 0.5 | 76.3 ± 0.2 | 63.8 ± 0.1 | 74.20 |
| ✓ | ✗ | ✓ | 72.1 ± 0.5 | 84.0 ± 2.3 | 76.7 ± 1.3 | 77.9 ± 0.7 | 82.5 ± 0.5 | 61.4 ± 0.3 | 76.3 ± 0.2 | 64.3 ± 0.7 | 74.40 |
| ✓ | ✓ | ✓ | 73.5 ± 0.9 | **84.9 ± 1.3** | **79.4 ± 0.9** | **78.8 ± 0.5** | **85.2 ± 0.4** | 61.2 ± 1.0 | **76.9 ± 0.1** | **64.9 ± 0.2** | **75.60** |

#### A.4.2 Impact of hyper-parameters of $\mathcal{L}_{T-soft}$.

Since $\mathcal{L}_{T-soft}$ is the most important loss component, we further analyze hyper-parameters associated with it. We can see in Eq. 9, $\mathcal{L}_{soft}$ is similar to the distillation loss for classification tasks, consisting of two loss components, $i.e.$ $\mathcal{L}_{KD}$ (Eq. 7) and $\mathcal{L}_{wGD}$ (Eq. 8). Here, we analyze the temperature term, $\tau$ for $\mathcal{L}_{KD}$, followed by the weights of these components, $\alpha$.

Table 10: Impact of the temperature, $\tau$. (at $\alpha = 0.95$)

| $\tau$ | BBBP | ClinTox | MUV | HIV | BACE | SIDER | Tox21 | ToxCast | $Avg$ |
|---|---|---|---|---|---|---|---|---|---|
| 1 | **75.1 ± 0.4** | **86.7 ± 1.6** | 74.4 ± 0.1 | 77.5 ± 0.6 | 83.3 ± 1.0 | 61.2 ± 0.4 | 75.6 ± 0.1 | 63.4 ± 0.4 | 74.65 |
| 5 | 73.4 ± 0.2 | 81.8 ± 1.4 | 77.3 ± 2.3 | 78.6 ± 0.6 | 83.8 ± 0.8 | 61.7 ± 0.8 | 76.5 ± 0.3 | 63.9 ± 0.4 | 74.63 |
| 10 | 73.5 ± 0.9 | 84.9 ± 1.3 | **79.4 ± 0.9** | **78.8 ± 0.5** | **85.2 ± 0.4** | 61.2 ± 1.0 | **76.9 ± 0.1** | **64.9 ± 0.2** | **75.60** |
| 20 | 72.9 ± 0.6 | 83.9 ± 2.6 | 77.1 ± 0.5 | 78.3 ± 0.7 | 84.0 ± 0.8 | 61.8 ± 0.4 | 76.2 ± 0.4 | 64.6 ± 0.5 | 74.85 |
| 100 | 73.6 ± 0.1 | 80.6 ± 0.2 | 76.8 ± 2.8 | 78.7 ± 1.1 | 84.0 ± 0.5 | **62.3 ± 0.5** | 76.1 ± 0.3 | 64.5 ± 0.2 | 74.58 |

In Table 10, we demonstrate the results by varying the temperature, $\tau$ to $\{1, 5, 10, 20, 100\}$. We observe that at a lower temperature of $\tau = 1$, we achieve the best performance for BBBP and Clintox datasets, while the performance remains lower for the other datasets. On the other hand, at $\tau = 100$, we achieve the best performance for SIDER. Finally, we obtain the most consistent result as we select $\tau = 10$ and set it to report our results.

Table 11: Impact of $\alpha$. (at $\tau = 10$)

| $\alpha$ | BBBP | ClinTox | MUV | HIV | BACE | SIDER | Tox21 | ToxCast | $Avg$ |
|---|---|---|---|---|---|---|---|---|---|
| 0 | 73.2 ± 0.7 | 80.2 ± 1.8 | 76.4 ± 0.7 | 78.1 ± 0.6 | 84.1 ± 0.9 | 62.3 ± 0.5 | 75.5 ± 0.3 | 63.8 ± 0.3 | 74.20 |
| 0.5 | **73.6 ± 0.7** | 82.5 ± 1.2 | 75.0 ± 1.5 | 78.4 ± 0.6 | **85.6 ± 0.5** | **62.4 ± 0.1** | 75.8 ± 0.1 | 64.5 ± 0.3 | 74.73 |
| 0.95 | 73.5 ± 0.9 | **84.9 ± 1.3** | **79.4 ± 0.9** | **78.8 ± 0.5** | 85.2 ± 0.4 | 61.2 ± 1.0 | **76.9 ± 0.1** | **64.9 ± 0.2** | **75.60** |
| 1.0 | 72.8 ± 0.6 | 83.6 ± 1.2 | 77.6 ± 1.6 | **78.8 ± 0.4** | 83.4 ± 1.3 | 61.3 ± 0.6 | 76.6 ± 0.3 | 64.1 ± 0.2 | 74.78 |

Next, in Table 11, we analyze the impact of $\alpha$ with fixed $\tau = 10$. A larger value of $\tau$ provides more weight to $\mathcal{L}_{KD}$. We can see that increasing $\alpha$ to a non-zero value improves the model's overall performance. However, performance tends to reduce as we choose $\alpha = 1$ to remove $\mathcal{L}_{wGD}$ entirely. We achieve the best performance at $\alpha = 0.95$.

Table 12: Sensitivity analysis of $\lambda_1$ and $\lambda_2$ for $TGCL - DSLA(w/DSLA)$ model.

|  | $\lambda_1 = 0.3$ | $\lambda_1 = 0.5$ | $\lambda_1 = 0.7$ | $\lambda_1 = 1.0$ |
|---|---|---|---|---|
| **BBBP** | $72.55 \pm 0.32$ | $72.03 \pm 0.59$ | $\mathbf{74.46 \pm 1.54}$ | $73.5 \pm 0.9$ |
| **ClinTox** | $81.60 \pm 0.21$ | $80.33 \pm 2.39$ | $82.95 \pm 0.75$ | $\mathbf{84.9 \pm 1.3}$ |
| **BACE** | $83.64 \pm 0.71$ | $83.18 \pm 1.02$ | $83.38 \pm 0.33$ | $\mathbf{85.2 \pm 0.4}$ |
| **MUV** | $76.28 \pm 0.91$ | $76.00 \pm 0.37$ | $77.08 \pm 1.45$ | $\mathbf{79.4 \pm 0.9}$ |
| **HIV** | $78.61 \pm 0.62$ | $\mathbf{78.93 \pm 0.58}$ | $78.64 \pm 0.46$ | $78.8 \pm 0.5$ |
|  | $\lambda_2 = 0.3$ | $\lambda_2 = 0.5$ | $\lambda_2 = 0.7$ | $\lambda_2 = 1.0$ |
| **BBBP** | $72.96 \pm 0.27$ | $73.5 \pm 0.9$ | $72.85 \pm 0.24$ | $\mathbf{74.12 \pm 0.41}$ |
| **ClinTox** | $83.53 \pm 1.51$ | $\mathbf{84.9 \pm 1.3}$ | $81.95 \pm 0.93$ | $80.83 \pm 1.68$ |
| **BACE** | $83.33 \pm 0.31$ | $\mathbf{85.2 \pm 0.4}$ | $83.60 \pm 0.90$ | $83.53 \pm 0.72$ |
| **MUV** | $75.79 \pm 0.78$ | $\mathbf{79.4 \pm 0.9}$ | $76.75 \pm 1.54$ | $76.02 \pm 0.11$ |
| **HIV** | $78.66 \pm 0.81$ | $78.8 \pm 0.5$ | $78.17 \pm 0.62$ | $\mathbf{79.49 \pm 0.84}$ |

### A.4.3 Sensitivity of $\lambda_1$ and $\lambda_2$

In Table 12, we present the performance of $TGCL - DSLA(w/DSLA)$ model as we vary $\lambda_1$ and $\lambda_2$ in Eq. 11. We first vary $\lambda_1$ to $\{0.3, 0.5, 0.7, 1.0\}$ as we fix $\lambda_2 = 0.5$. We observe that the performance improves as we choose larger values, *i.e.*, when we set $\lambda_1$ to 0.7 or 1.0. Next, we vary $\lambda_2$ to $\{0.3, 0.5, 0.7, 1.0\}$ as we fix $\lambda_1 = 1.0$. Here, we observe that we achieve the average performance as we set $\lambda_2 = 0.5$

