# OpenReview forum: "Teacher-Guided Graph Contrastive Learning"
_TMLR — Accepted by TMLR_

### Review · Reviewer_ydtR · 2024-08-25

**Summary Of Contributions:**

This paper proposes Teacher-Guided Graph Contrastive Learning (TGCL), a framework for self-supervised representation learning on graphs. TGCL uses a pre-trained "teacher" model to guide a "student" model's training by providing soft pseudo-labels based on perceptual distances between graph samples. Two variations are presented: one based on NT-Xent loss and another on the D-SLA method. Experiments on molecular property prediction and social network link prediction tasks are conducted to evaluate the approach.

**Audience:**

No

**Claims And Evidence:**

Yes

**Requested Changes:**

Comments and Suggestions:

- The computational overhead of the teacher-student framework should be quantified and discussed.
- A detailed analysis of how student network performance varies with teacher capacity is required.
- The paper should discuss potential limitations and failure cases of the proposed approach more thoroughly.

Minor comments:

- Figure 2 panel b D-SLA uses the same schema for attraction and repulsion
- In section 3.2.1, "pseudo-level" is used - should this be "pseudo-label"?
- Equation 5 has extra brackets

**Strengths And Weaknesses:**

Strengths:

1. Empirical results: The proposed methods show improvements over baselines on multiple benchmark datasets. The paper provides ablation studies and analysis of different components.

Weaknesses:

1. Computational overhead: The teacher-student framework introduces additional computational cost compared to single-model approaches. A detailed account of this overhead is lacking.
2. Dependence on teacher quality: The performance gains rely on having a good pre-trained teacher model. The paper lacks a thorough analysis of how student network performance varies with teacher capacity.
3. Limited novelty: While the teacher-student framework is applied to graphs, the core idea is borrowed from existing knowledge distillation literature.

Despite presenting an interesting approach to graph representation learning, this paper falls short of making a significant contribution. The marginal performance gains do not justify the added complexity and computational cost of the teacher-student framework. The lack of thorough analysis on computational overhead and teacher-student dynamics, combined with limited novelty, weakens the paper's impact. Unless these issues are substantially addressed, rejection is recommended.

---

> ### Author Response · Authors · 2024-09-07
> **Response to Reviewer ydtR**
>
> We thank reviewer ydtR for their constructive feedback. Please find below our inline response to the queries raised.
>
> **Computational overhead: The teacher-student framework introduces additional computational cost compared to single-model approaches. A detailed account of this overhead is lacking.**
>
> We have added additional experiments in Appendix A.3.3 to address these concerns. Overall in Appendix A.3, we evaluate the performance of TGCL models with various teacher models and model capacities. We also analyze the trade-off between computational overhead and convergence by applying undertrained teacher models vs. a well-trained teacher model.
>
>
> **Dependence on teacher quality: The performance gains rely on having a good pre-trained teacher model. The paper lacks a thorough analysis of how student network performance varies with teacher capacity.**
>
> In Appendix A.3.3, we present the performance of our student models using undertrained teacher models. Our results demonstrate that the student models, even under the guidance of undertrained teachers, consistently outperform the fully-trained teacher. This indicates that the performance of the student model does not entirely depend on having a well-trained teacher.
> Additionally, Appendix A.3.1 presents results for TGCL student models with varying capacities relative to the teacher network. We observe that reducing the capacity of the student network by decreasing the number of layers generally leads to a decline in performance. Nevertheless, even with a student model consisting of just 3 layers of GNN, our student module still outperforms the teacher D-SLA model.
>
> **Limited novelty: While the teacher-student framework is applied to graphs, the core idea is borrowed from existing knowledge distillation literature.**
>
> Yes, we agree that the core idea is borrowed from existing knowledge distillation literature. However, we have proposed novel loss functions, specifically for discrete structures of graphs to incorporate the concept of distillation in the CL framework. To the best of our knowledge, we are the first to propose a teacher-guided soft-discrimination-based contrastive learning framework for graphs.
>
>
> **The paper should discuss potential limitations and failure cases of the proposed approach more thoroughly.**
>
> We have provided an additional discussion on potential limitations and also advantages of the framework. Please refer to section 5: Conclusion & Discussion. Further, we provide an additional set of empirical results to investigate these challenges (Appendix A.3).

---

### Review · Reviewer_QeW5 · 2024-08-27

**Summary Of Contributions:**

This paper proposes a novel contrastive learning framework for GNNs by incorporating more semantically-rich soft-discriminative features using pretrained teacher. Specifically it proposes teacher-guided distilled distance which is designed to accommodate the discrete nature of graphs via knowledge distillation.

**Audience:**

Yes

**Claims And Evidence:**

Yes

**Requested Changes:**

1. Authors may add more interpretation insight on how the KD helps the CL.
2. It seems that in this paper, the teacher model is not nesseary to be larger than the student models. It can be pretrained peer GNN as well. Can you elaborate more on that?

**Strengths And Weaknesses:**

Strengths
1. The idea is novel. Authors incorporate the knowledge distillation method into graph contrastive learning by injecting soft pseudo-labels into contrastive loss objective.
2. The general KD idea is flexible to applied into any CL methods including NTXent and DSLA.

Weaknesses:
1. It is not clear why authors combine the student distance and teacher's distilled distance by D_dp(G_0, G_i) f(G_0) f(G_i). Why not D_dp(G_0, G_i) || f(G_0) - f(G_i)|| _2?
2. The TGCL performance does not have a clear improvement for DSLA method shown in Table 2.
3. More interpretation on how KD help those generalization in applications are of interest, while we only see that in Figure 1

---

> ### Author Response · Authors · 2024-09-07
> **Response to Reviewer QeW5**
>
> We thank reviewer QeW5 for their appreciation of our work and for their constructive feedback. Below, we provide our detailed responses to the queries raised.
>
> **It is not clear why authors combine the student distance and teacher's distilled distance by D_dp(G_0, G_i) f(G_0) f(G_i). Why not D_dp(G_0, G_i) || f(G_0) - f(G_i)|| _2?**
>
> Yes. One can also explore a variant formulation using $L_2$ similarity-based variation to define the loss function as well. We have highlighted this point in Section 3.3.1. In general, the idea is to multiply the similarities i.e., $sim(G_0, G_i)$ with $D_{dp}(G_0, G_i)$. Since the NT-XEnt formulation uses $sim(G_0, G_i) = f(G_0)\cdot f(G_i)$ i.e., as the dot product, we also incorporate $D_{dp}(G_0, G_i)$ upon the same formulation.
>
>
> **Authors may add more interpretation insight on how the KD helps the CL.**
>
> We have included an additional discussion on potential limitations as well as advantages of a CL framework with KD. Please refer to section 5: Conclusion & Discussion. Further, we provide an additional set of empirical results to investigate these challenges (Appendix A.3).
>
> **It seems that in this paper, the teacher model is not necessary to be larger than the student models. It can be pretrained peer GNN as well. Can you elaborate more on that?**
>
> Yes. We have included additional discussions in Appendix A.3.1 to elaborate on the fact that the key results in our main paper use peer GNN for the student. Also, we provide a set of empirical results on the choice of teacher network, capacity of the student models, and knowledge saturation phenomenon for our TGCL framework.

---

### Review · Reviewer_N9yF · 2024-09-01

**Summary Of Contributions:**

This paper proposes a teacher-student framework for improving the existing contrastive learning methods for graphs. The proposed method leverages a teacher model to produce soft pseudo-labels for pairs of graphs, which help the student model to achieve better generalization via knowledge distillation. Experiments on biological and social network datasets demonstrate performance improvement over some state-of-the-art methods.

**Audience:**

Yes

**Claims And Evidence:**

Yes

**Requested Changes:**

Proposition 1 should be rewritten to be more precise. Is it possible to develop a stronger and more direct theoretical guarantee for the proposed method?

Minor issue: The AUROC scores reported in all tables seem to have been multiplied by 100. This should be mentioned somewhere in the text.

**Strengths And Weaknesses:**

Strengths:

The paper correctly points out the limitations of using hard labels for positive and negative pairs in graph contrastive learning, which provides a strong motivation for the methodology. The experiments seem to be carefully designed and conducted.

Weaknesses:

Although Section 3.2 provides some justifications for the ideas behind the proposed method, direct theoretical guarantees are still lacking. In particular,
1) The statement of Proposition 1 is vague. What does the equivalence exactly mean? I cannot find the relevant discussion in Section 2.1 of Gutman and Hyvarinen (2010).
2) Proposition 2 justifies the superiority of Bayes-distilled risk in the context of supervised learning. However, without a precise equivalence between graph contrastive learning and supervised learning, it does not lead to a formal theoretical guarantee for the proposed framework.

---

> ### Author Response · Authors · 2024-09-07
> **Response to Reviewer N9yF**
>
> We thank reviewer N9yF for their constructive feedback. Please find below our inline response to the queries.
>
> **Proposition 1 should be rewritten to be more precise. Is it possible to develop a stronger and more direct theoretical guarantee for the proposed method?**
>
> Noise-contrastive estimation (NCE) estimates the true probability density function of a random variable by training a binary classifier to distinguish between samples from the true distribution and samples from a noise distribution. The estimation of the true density function is derived from the learned binary classification function.
>
> - Please refer to Section 2 in Gutmann & Hyvärinen (2010) (not Section 2.1 alone). In particular, Eq. 3 and Eq. 6-11 describe this setting.
> - We have rewritten Proposition 1 and the corresponding text to make it clearer and precise (highlighted in Blue).
>
>
>
> **Minor issue: The AUROC scores reported in all tables seem to have been multiplied by 100. This should be mentioned somewhere in the text.**
>
> We have updated it in the captions of Tables 1, 2, and 3.

---

### Author Response · Authors · 2024-09-07
**Highlights of edits based on reviewers' feedback**

We sincerely thank the Action Editor for handling the review process of our work and the reviewers for their insightful reviews of our manuscript that helped us to significantly improve it. We have highlighted the edited parts of our revised manuscript using blue color. We have tried our best to address all the concerns raised by the reviewers. We provide a point-by-point response to the issues raised in our response. Below is a summary of the various significant edits:

- We have rewritten proposition 1 to improve its clarity and precision (Section 3.2).
- We have edited Section 5 to include more interpretation insight on how the KD helps the CL and also elaborate on potential challanges of our framework.
- We have performed additional experiments and reported in Appendix A.3.3 to study the impact of teacher’s quality on the student network’s performance.
- We have added a discussion in Appendix A.3.1. to elaborate more on the fact that the teacher network need not be larger than the student network.

---

### Decision · Action_Editor_aJZS · 2024-11-06

**Recommendation:** Accept with minor revision

**Comment:**

The paper introduces a teacher-student framework for graph contrastive learning, leveraging a pre-trained teacher to generate soft pseudo-labels for improved generalization. Empirical results across benchmark datasets support the approach's effectiveness, showing notable improvements over traditional hard label-based methods. Reviewers highlighted the paper's robust empirical performance, practical value, and the flexibility to integrate with existing contrastive methods. Concerns were raised about computational overhead, reliance on teacher quality, and limited theoretical justification, particularly regarding the clarity of Propositions 1 and 2. However, the authors addressed these concerns during the response period, clarifying aspects of the computational overhead and enhancing the justification of their approach. Given its innovative contributions and practical relevance, I recommend acceptance.

**Audience:**

Yes, the paper’s innovative approach to enhancing graph contrastive learning through a teacher-student framework would be of interest to TMLR’s audience, particularly those focused on representation learning, graph neural networks, and those related applications.

**Claims And Evidence:**

The paper presents a novel teacher-student framework for graph contrastive learning, yielding promising empirical results across multiple datasets and demonstrating careful experimental design. The proposed method leverages a teacher model to produce soft pseudo-labels for pairs of graphs, which help the student model to achieve better generalization via knowledge distillation. However, reviewers note areas for improvement, such as clarifying computational overhead and strengthening theoretical backing. Overall, the innovative approach and empirical success support acceptance with minor revisions. The authors are strongly encouraged to further improve the theoretical guarantee and the com